# Population genomic signatures of the oriental fruit moth related to the Pleistocene climates

Li-Jun Cao[1], Wei Song[1,2], Jin-Cui Chen[1], Xu-Lei Fan[1], Ary Anthony Hoffmann[3] & Shu-Jun Wei [1✉]

The Quaternary climatic oscillations are expected to have had strong impacts on the evolution of species. Although legacies of the Quaternary climates on population processes have been widely identified in diverse groups of species, adaptive genetic changes shaped during the Quaternary have been harder to decipher. Here, we assembled a chromosome-level genome of the oriental fruit moth and compared genomic variation among refugial and colonized populations of this species that diverged in the Pleistocene. High genomic diversity was maintained in refugial populations. Demographic analysis showed that the effective population size of refugial populations declined during the penultimate glacial maximum (PGM) but remained stable during the last glacial maximum (LGM), indicating a strong impact of the PGM rather than the LGM on this pest species. Genome scans identified one chromosomal inversion and a mutation of the circadian gene *Clk* on the neo-Z chromosome potentially related to the endemicity of a refugial population. In the colonized populations, genes in pathways of energy metabolism and wing development showed signatures of selection. These different genomic signatures of refugial and colonized populations point to multiple impacts of Quaternary climates on adaptation in an extant species.

[1] Institute of Plant Protection, Beijing Academy of Agriculture and Forestry Sciences, 9 Shuguanghuayuan Middle Road, Haidian District, Beijing 100097, China. [2] Beijing Key Laboratory for Forest Pests Control, Beijing Forestry University, Beijing 100083, China. [3] School of BioSciences, Bio21 Institute, University of Melbourne, Parkville, VIC, Australia. ✉email: shujun268@163.com

In the Quaternary, species are expected to have contracted to refugial areas during periods of glaciation and then expanded their distribution during warm interglacial periods[1]. These climatic oscillations would have had extensive impacts on the evolution of species[2,3]. Understanding the genetic legacy of species contractions and expansions during such periods of extreme climate changes is a fundamental goal of molecular evolutionary studies[4,5].

The effects of Quaternary climate changes on species evolution are expected to have varied with the geographical range of species and period of glaciation. In Europe and North America, the last glacial maximum (LGM, ∼26.5–19 thousand years ago, kya)[6] would have had a strong impact on species, given that many species were confined to refugial area in the LGM and colonized their current distribution range after this period[7,8], and with genetic legacies of Quaternary climates expected to depend on this period[3,7,9]. In contrast, climate oscillations have been mild in East Asia during the LGM. Many species spread out from their refugia early before the penultimate glacial maximum (PGM, ∼155–140 kya)[10,11]. Consequently, species in East Asia are expected to have experienced different climatic changes in the Pleistocene with different genetic legacies. However, studies on genetic legacies in species from this area lag those in Europe and North America[11,12].

Under climate changes, neutral and adaptive variations can be reshaped in different populations[1,13,14]. During the glacial and the postglacial dispersal periods, populations would have been isolated with gene flow restricted—genetic differentiation at neutral loci would have developed due to genetic drift in small refugial populations[15]. Neutral genetic variation has been widely investigated across populations to build an ecological and evolutionary understanding of species during periods of change[8,16–19]. In addition, changing climates pose selection pressures on populations that can respond to local climates by adaptive evolution[14,20,21]. While adaptive evolution to climate changes has been identified across many species[14], populations used for genome scans of climate change adaptation have tended to focus on more recent historical periods of climate change in Europe, North America, and Australia[22–25]. We still lack an understanding of the adaptive responses of species to climatic changes since the early Pleistocene. Nevertheless, insights into the adaptive evolution of species to past environments may improve our understanding of species' responses to ongoing and future climate changes[14,26,27].

The Tortricidae represent one of the largest families of Lepidoptera (Hexapoda: Insecta) with over 5000 described species, including many economically important pests in agriculture and forestry[28]. Moths of the family Tortricidae tend to distribute in temperate and tropical upland regions[28]. However, some species have expanded their distribution range to become global invasive species[29,30]. The oriental fruit moth, *Grapholita molesta*, representing one such species, has become a cosmopolitan pest of stone and pome fruits[31]. This species is native to East Asia and molecular data suggest two refugial regions around eastern Qinghai-Tibet Plateau boundary regions (Yunnan and Sichuan Basin)[12,32]. This moth has dispersed to temperate and tropical upland regions of all continents within the last 100 years[30]. Genetic lineages from one of the refugial areas (Yunnan) appear to have expanded post-glacially to the east and then to the north of China, while genetic lineages from another refugial area (Sichuan Basin) are considered endemic[12,32]. The native populations of the oriental fruit moth provide ideal models to understand adaptive genomic variation related to climate changes of the Quaternary in East Asia.

In this study, we assembled a high-quality chromosome-level genome for the oriental fruit moth and examined genomic signatures potentially related to the Quaternary climate changes in the refugial and colonized populations. We found that refugial populations of the oriental fruit moth were mainly impacted by the PGM but not the LGM. Higher differentiation on the neo-Z chromosome was detected in the endemic refugial populations. Genomic scans identified a circadian gene that was related to the endemicity of the refugial populations, and wing development genes that were related to the colonized populations. These results indicated that both the extreme climate during the glaciation and the dispersal event in the postglacial periods might shape the genomic signatures of the oriental fruit moth in East Asia. As far as we know, this is the first attempt to identify adaptive genomic variations among populations that diverged in the early Pleistocene period of the Quaternary.

## Results

**A high-quality chromosome-level genome for the oriental fruit moth.** We generated 53.7 Gb NanoPore long-reads, 48.7 Gb Illumina short-reads, and 120.8 Gb Hi-C short-reads for the oriental fruit moth genome assembly (Supplementary Table 1). The estimated size of the oriental fruit moth genome ranged from 393 to 439 Mb, heterozygosity ranged from 0.721 to 0.865%, and duplication rate ranged from 1.55 to 1.73% when the k-mer value was varied from 17 to 31 (Supplementary Fig. 1). The assembled contigs covered 521.6 Mb in size with an N50 of 1.79 Mb. These contigs were further assembled into 28 chromosomes with a total length of 517.71 Mb and a BUSCO completeness of up to 97.1% (Fig. 1e, Table 1). In total, we annotated 15,269 protein-coding genes (Table 1), 61 8s_rRNAs, 63 5s_rRNAs, and 29786 tRNAs (Supplementary Table 2), as well as 15.37 Mb repetitive elements (11.01% of the genome) in this genome (Supplementary Table 3).

To estimate the reliability of the genome assembly, we conducted a synteny analysis between the oriental fruit moth, the species *Cydia pomonella* which represents another chromosome-level genome of a Tortricidae[29], and *Spodoptera litura*. Strong syntenic blocks were found between the oriental fruit moth, *C. pomonella* and *S. litura* genomes (Fig. 1f). Each chromosome showed one-to-one synteny between the oriental fruit moth and *C. pomonella* genomes, except for the sex chromosome W (chromosome 29 in *C. pomonella*), which was not assembled in the oriental fruit moth genome. In the Tortricidae, the number of chromosomes ranges from 27 to 31[33]. The subfamily Tortricinae has 30 chromosomes, while the Olethreutinae (including the oriental fruit moth and *C. pomonella*) has 28 chromosomes[28]. Three fusion events were identified in the tortricid moths. A neo-Z chromosome was formed by a fusion event between the sex chromosome Z and an autosome[29,34,35]. The second fusion involved chromosome 7 and chromosome 8 of *S. litura* fused into chromosome 2 of the oriental fruit moth. The third one involved the fusion of chromosome 5 and chromosome 22 of *S. litura* into chromosome 4 of the oriental fruit moth (Fig. 1f). The genome comparisons indicate that the chromosome number and fusion events are conserved in the two sequenced tortrix genomes.

To explore possible genomic components related to environmental adaption in the oriental fruit moth as well as the other tortricid moth, *C. pomonella*, we manually annotated detoxification genes, chemosensory genes, and heat shock proteins (HSP) genes and compared these gene families among 13 representative genomes of Lepidoptera, as described in Supplementary Table 4. The two tortricid moths have a moderate number of detoxification genes, and the lowest number of HSP genes, of the Lepidoptera characterized so far. In contrast, the oriental fruit moth has the highest number of P450s, IRs, and ORs. Compared to *C. pomonella*, the oriental fruit moth has more genes in five annotated gene families (Supplementary Table 4).

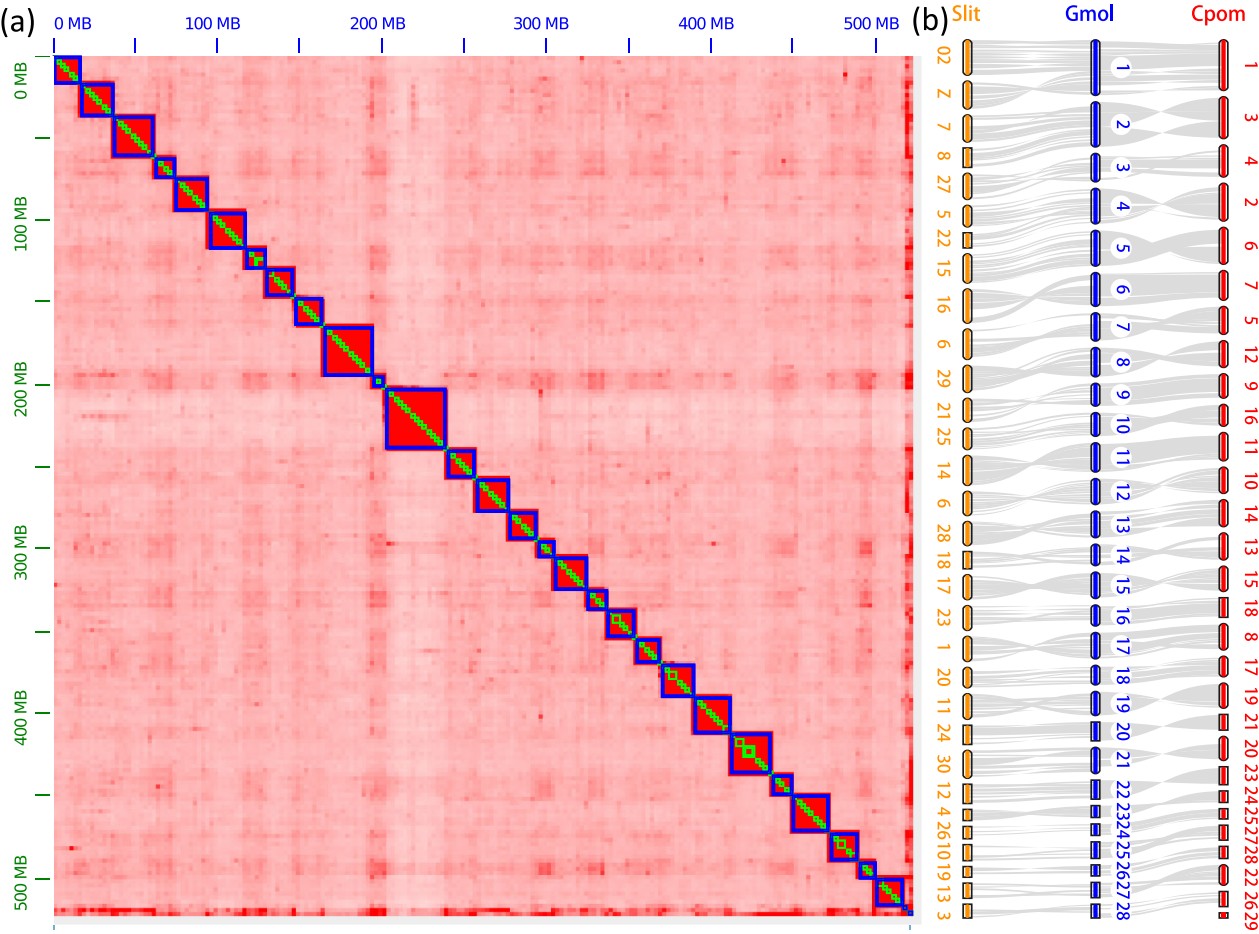

**Fig. 1 Genome assembly of the oriental fruit moth, *Grapholita molesta*. a** The genome-wide all-by-all Hi-C matrix of OFM. The genome sequences were assembled using Juicer grouped using 3D-DNA; 28 linkage groups were identified based on Hi-C contact, indicated by blue boxes. Only sequences anchored on chromosomes are shown in the plot. **b** Synteny blocks among *Grapholita molesta* (Gmol), *Spodoptera litura* (Slit), and *Cydia pomonella* (Cpom) genomes. The orange, blue and red oval blocks represent chromosomes assembled in *S. litura*, *G. molesta* and *C. pomonella*; the gray lines represent the synteny blocks between the genomes. Strong synteny blocks were identified between the three moth genomes.

**Genetic diversity among populations of the oriental fruit moth**. The high-quality genome assembly provides an opportunity to examine chromosome evolution and genomic signatures potentially related to the Quaternary climate changes in the refugial and colonized populations (Fig. 2). We called 6,338,975 SNPs from three populations of the oriental fruit moth; these were from the original and refugial area in Yunnan (YNHH), from another refugial area in Sichuan (SCCD), and from a colonized area of northern China (BJPG) (Supplementary Table 5, Fig. 3).

Because neo-Z chromosomes have been reported as increasing adaptive potential in tortricid pests[34], we wondered if the neo-Z chromosome had different characteristics to the autosomes. We examined population genetic diversity and differentiation across chromosomes before scanning the genome for outliers. Values obtained from sliding windows of chromosomes were considered to be significantly different if their 99% confidence intervals (mean ± 2.58 × std.err) did not overlap. The neo-Z chromosome had the highest $\pi$ and $F_{ST}$ values compared with the autosomes when calculated for all population pairs. The average $\pi$ value for the neo-Z chromosome (0.00327 ± 0.00006) was ~1.56 fold that of the autosomes (ranging from 0.00110 ± 0.00004 to 0.00279 ± 0.00009 with an average of 0.02094 ± 0.00006). Average $F_{ST}$ values of the neo-Z chromosome (0.2735 ± 0.0041, 0.2830 ± 0.0042 and 0.2055 ± 0.0043) were 1.82, 1.74, and 1.40 fold higher than for the autosomes (0.1502 ± 0.0008, 0.1630 ± 0.0008, and 0.1466 ± 0.0008)

for population pairs of SCCD vs. BJPG, SCCD vs. YNHH, and BJPG vs. YNHH, respectively (Supplementary Fig. 2). This higher differentiation may be related to a lower effective population size or reduced gene flow effects on the neo-Z chromosome.

To test for expected differences in demographic changes in the three populations of the oriental fruit moth, we compared their genetic diversity and inferred demographic history based on genome resequencing data. The SCCD had a larger estimated genome size, heterozygosity, and duplication rate than the recently colonized BJPG population (Supplementary Fig. 3). The refugial SCCD population had the highest genetic diversity and a higher proportion of rare alleles (Supplementary Figs. 4–5). In contrast, the other two populations had similar genetic diversity in terms of nucleotide diversity (Supplementary Figs. 4–5). The SCCD population also had a lower linkage disequilibrium (LD) level than the other two populations, with LD decaying to its half-maximum within 0.3 kb, 1.8 kb, and 2.7 kb in SCCD, BJPG, and YNHH, respectively (Supplementary Fig. 6). These results suggest that the YNHH and BJPG populations may have been exposed to stronger bottlenecks than the SCCD population, as revealed by smc++ analysis. Pairwise $F_{ST}$ values between populations were 0.1517 ± 0.0009 (BJPG vs. SCCD), 0.1480 ± 0.0008 (BJPG vs. YNHH), and 0.1655 ± 0.0009 (SCCD vs. YNHH), indicating that the SCCD population is particularly strongly differentiated from the YNHH population.

**Table 1 Completeness of major versions of assembled and annotated genome of Grapholita molesta estimated by BUSCO.**

| Step | Version | Software | Complete % | Single copy % | Duplication % | Fragment % | Missing % | Size (Mb)/No. gene | N50 (Mb) |
|---|---|---|---|---|---|---|---|---|---|
| Assembly | Contig assembly | CANU | 83.3 | 57.7 | 25.6 | 7.1 | 9.6 | 949.05 | 0.85 |
| | Synteny reduction | purge_haplogs | 76.6 | 74.7 | 1.9 | 10.5 | 12.9 | 521.60 | 1.79 |
| | Contig polish | PILON | 96.7 | 93.1 | 3.6 | 1.1 | 2.2 | 524.93 | 1.79 |
| | Linkage group assembly | Juicer + 3D-DNA | 93.3 | 92.2 | 1.1 | 1.0 | 5.7 | 525.22 | 19.5 |
| | Linkage group polish | PILON | 97.1 | 96.1 | 1.0 | 1.0 | 1.9 | 517.71 | 19.5 |
| Annotation | Annotation | MAKER | 90.1 | 86.1 | 4.0 | 6.6 | 3.3 | 39,274 | NA |
| | Filtering based on expression | In-home script | 89.8 | 85.9 | 3.9 | 6.1 | 4.1 | 19,478 | NA |
| | Gene structure improvement | PASA | 92.6 | 76.6 | 16.0 | 4.4 | 3.0 | 19,968 | NA |
| | Functional annotation | EGGNOG | 92.6 | 76.6 | 16.0 | 4.3 | 3.1 | 15,269 | NA |

NA not available.

**Demographic history under climate changes**. All populations experienced a decline in effective size that started at ~800 kya and ended at ~115 kya in the YNHH and SCCD refugial populations and the BJYQ colonized population (Fig. 4a). This period coincided with the Middle Pleistocene (781–126 kya) and PGM (155–140 kya), and is consistent with the divergence time of the Yunnan and Sichuan populations of the oriental fruit moth estimated from mitochondrial genes[12,32]. During the LGM, the effective population size of BJPG declined, while sizes of YNHH and SCCD were stable (Fig. 4a). These results suggest that the refugial YNHH and SCCD populations may be impacted by climates of the PGM but not that of LGM. After the LGM, the SCCD population began to expand. The BJPG and YNHH populations likely experienced bottlenecks after the LGM, with a particularly large decline in the BJPG population and then expansion in the last 10–300 years (Fig. 4a).

An analysis of divergence times showed that the SCCD population split earlier from the other populations, while the BJPG and YNHH populations split more recently from each other (Fig. 4b). This is consistent with variation in the effective size of the three populations after the LGM (Fig. 4a), and is also in agreement with the previous research[12,32,36].

**Genome structure variation among populations of the oriental fruit moth**. Structural variation in the oriental fruit moth genome was detected using a local PCA analysis (Fig. 5). Among the three populations, we found two regions that point to the presence of chromosomal inversions. A 160-Kbp small inversion occurred on chr5 covering five genes (Fig. 5a, Supplementary Data 1), while a larger 2.29-Mbp inversion occurred on chr24 covering 62 genes (Fig. 5b, Supplementary Data 1). We identified genotypes of all individuals for the inverted regions by plotting the observed heterozygosity and the first component of the PCA analysis on SNPs in the inverted regions (PC1). All individuals were separated into three clusters corresponding to the three genotypes. The heterozygous genotype had higher observed heterozygosity as might be expected (Fig. 5c, d). The small inversion is prevalent in the SCCD population, with nine out of 11 individuals possessing a homozygous inverted genotype (Fig. 5c). In comparison, the large inversion rarely occurred, with one individual from BJPG homozygous for the inversion, while one individual from BJPG, and two individuals from SCCD were heterozygous (Fig. 5d). When we removed individuals heterozygous and homozygous for the inversions, the LD in the remaining individuals across the region covered by the inversion was reduced to an average value (lower triangle in Fig. 5e, f).

**Genes potentially under selection in the refugial and colonized populations**. In testing the endemicity of the SCCD populations, 81 windows (50 kbp in size, green point in Fig. 6a, c) were filtered out as outlier regions in which SCCD individuals had lower diversity and higher differentiation than those from the other two populations. In these windows, 128 missense SNPs (maf > 0.05) have higher differentiation when comparing the SCCD population to others. PCAs of all SNPs separated individuals by populations (Fig. 6d), while PCAs of these outlier SNPs separated SCCD populations (Fig. 6e). In total, 42 genes were identified nearby these SNPs, 12 of which were located on the neo-Z chromosome (Supplementary Data 2). These genes may relate to the endemicity of SCCD. In addition, four outlier genes were located in the small inverted region of chr5 (Supplementary Data 1). One of the 12 outlier genes on the neo-Z chromosome is Clk, a core gene of the circadian clock. Low genetic polymorphism around Clk genes may limit adaptation to changed environmental conditions as suggested in other species[37,38]. Two

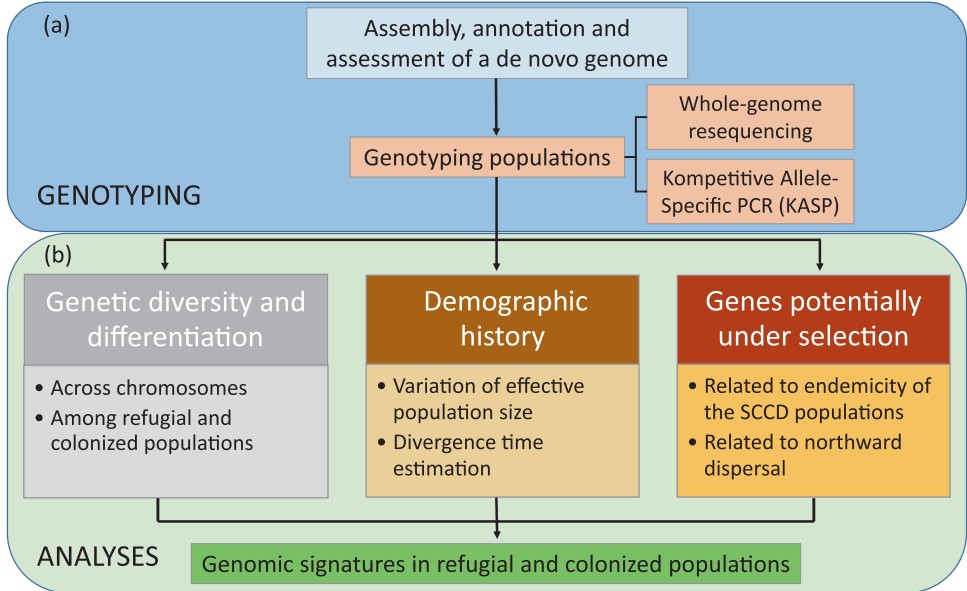

**Fig. 2 Overview of approaches to investigate genomic signatures of the oriental fruit moth related to the Pleistocene climates. a** Genotyping. Populations were genotyped using whole-genome resequencing and KASP methods based on a de novo assembled reference genome. **b** Data analysis. Genes potentially under selection were identified following genetic diversity, differentiation, and demographic history analyses.

of the 12 outlier genes on the neo-Z chromosome are NADH-ubiquinone oxidoreductase (Ndufb7) and an ABC transmembrane transporter, both involved in target-site based resistance of many pesticides[34]. Two SNPs related to the endemicity of the SCCD population were genotyped across 12 populations (Supplementary Data 3). The missense mutation on the Clk gene could only be found in two populations from the Sichuan Basin (Fig. 3a). The minor allele of the ABC gene was common in all four populations from the Sichuan Basin, while the reference allele was common in populations outside this basin (Fig. 3b).

For detecting outliers involved in northward dispersal, we obtained 138 outlier windows (50 kbp in size, red circles in Fig. 6b, c). The BJPG individuals showed lower diversity and higher differentiation than the refugial populations (SCCD and YNHH). A total of 149 missense SNPs (maf > 0.05) had higher differentiation in these windows between BJPG and the refuge populations than within the refuge populations. PCAs of these SNPs separated the BJPG population from the refuge populations and showed more dispersion in the refuge populations (Fig. 6f). In total, 54 genes were identified nearby these SNPs, one of which is located on the neo-Z chromosome (Supplementary Data 2). These genes were enriched for the amino sugar and nucleotide sugar metabolism pathways (Supplementary Fig. 7), which are involved in developing wings in another moth[39]. Two genes were annotated as involved in the carbohydrate metabolic process using eggNOG-Mapper. The orthologs of four genes have been reported in wing development, including two Cht3 genes[40–43] and two Serpin genes[44]. These genes may relate to selective sweeps during northward dispersal by the oriental fruit moth. Four SNPs were successfully genotyped across 12 populations (Supplementary Data 3), including one SNP marking the SRPN17 gene (Fig. 3c), two SNPs marking the Cht3 gene (Fig. 3d, e), and one SNP marking a gene involved in the carbohydrate metabolic process (Fig. 3f). The allele frequencies of these four SNPs revealed a similar pattern, distinguishing two refugial populations from those where dispersal has led to their establishment (Fig. 3c–f).

**Discussion**

In this study, we assembled a chromosome-level high-quality genome for the oriental fruit moth. We identified adaptive genomic signatures among native populations for the refugial and colonized populations of the oriental fruit moth that are potentially related to climate changes and range expansion during the Quaternary.

Several theoretical and empirical studies suggest that evolution may occur more rapidly on sex chromosomes[45,46]. In general, fewer recombination events and higher rates of drift are expected under a neutral process in Z or X chromosomes. Thus, the genetic diversity is lower in sex chromosomes than autosomes[47–49]. The relative diversity of Z or X chromosomes to autosomes ($\pi$Z/$\pi$A) is predicted to be 0.75[48]. The $\pi$Z/$\pi$A ratio is usually used to predict evolutionary history based on their deviation from the expectation of 0.75, involving processes such as population size fluctuations, sex breeding system and selection, speciation, and introgression, as documented in the human, fruit fly, and silk moth[47,49–51]. In the oriental fruit moth genome, we identified a neo-Z chromosome, which originated from the fusion of ancestral Z and autosomes, as in other tortricid moths[34]. The gene content and arrangement of the ancestral Z part of the oriental fruit moth was highly concordant with the Z chromosome of *S. litura* (Fig. 1), suggesting Z chromosome conservation across Lepidoptera[52]. In contrast to sex chromosomes of most species, we found significantly higher genetic diversity on the neo-Z chromosome than the autosomes of the oriental fruit moth ($p < 0.01$, 1.56 fold difference), reflecting a substantial departure from a 0.75 expectation. This high genetic diversity may increase the adaptive potential of the oriental fruit moth to different environments during its dispersal and invasion[34,53]. We found particularly high differentiation on the neo-Z chromosome in the SCCD population, and a high proportion of outlier genes related to SCCD endemicity detected on the neo-Z chromosome, suggesting that this chromosome plays an important role in adaptation of the SCCD population.

Lower genetic diversity might be expected in populations that are recently established due to bottlenecks associated with founder events and perhaps selective processes that favor strong dispersal/fast reproduction and lead to selective sweeps[4,5,29]. Consistent with this expectation, we detected a dramatic bottleneck reducing variation in a derived northern population of the oriental fruit moth, a moderate bottleneck in a refugial population

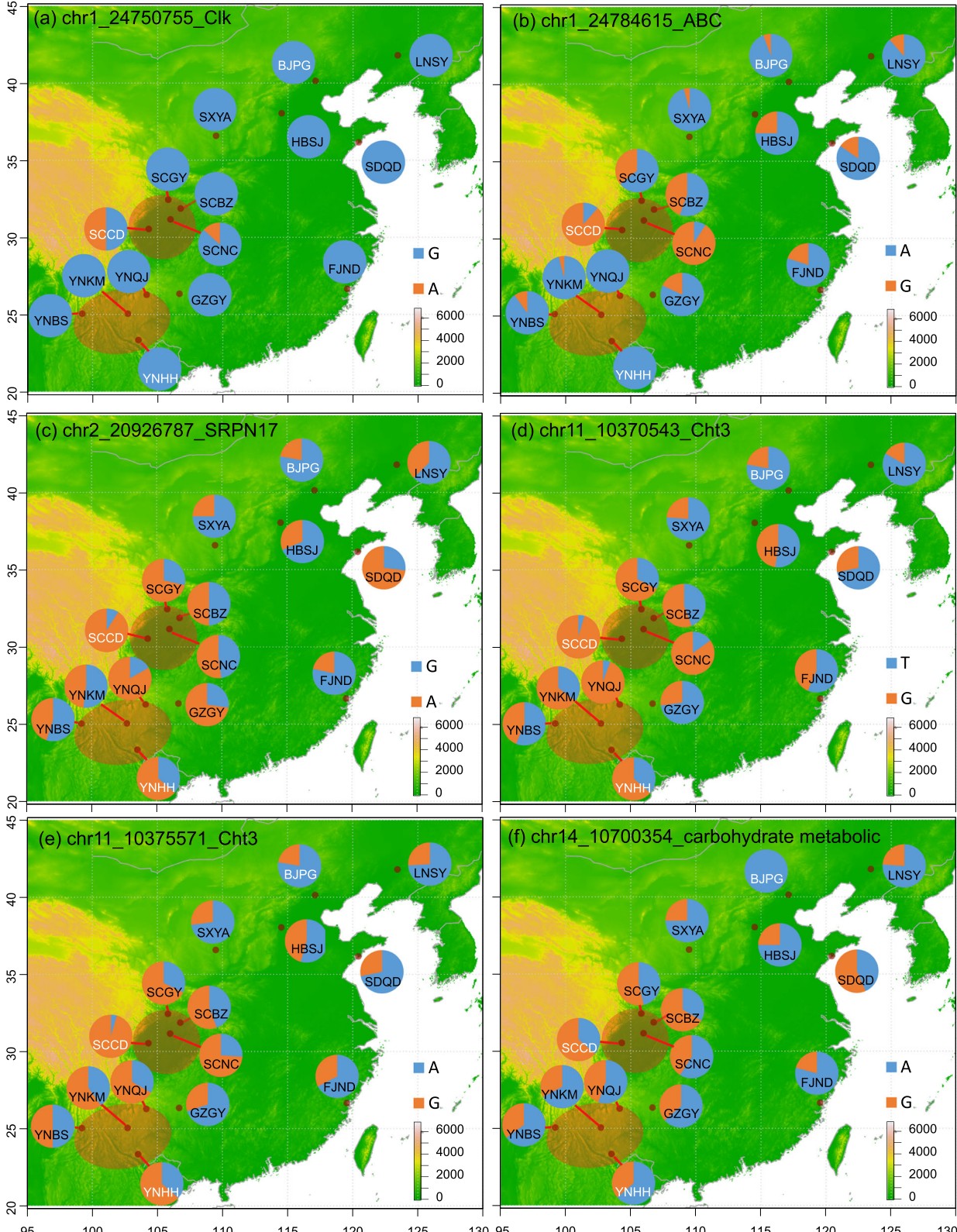

**Fig. 3 Location and allele frequency of the fifteen populations of the oriental fruit moth *Grapholita molesta* used in the study.** The subtitles represent chromosome ID, the position of SNP, and gene name or function. Population code in white letters was genotyped by resequencing, while black ones were genotyped by KASP (Kompetitive Allele-Specific PCR). The color of pie charts indicates the allele frequency of nucleotide of candidate SNPs across 15 populations. **a–b** SNPs putatively involved in endemicity; **c–f** SNPs putatively involved in northward dispersal.

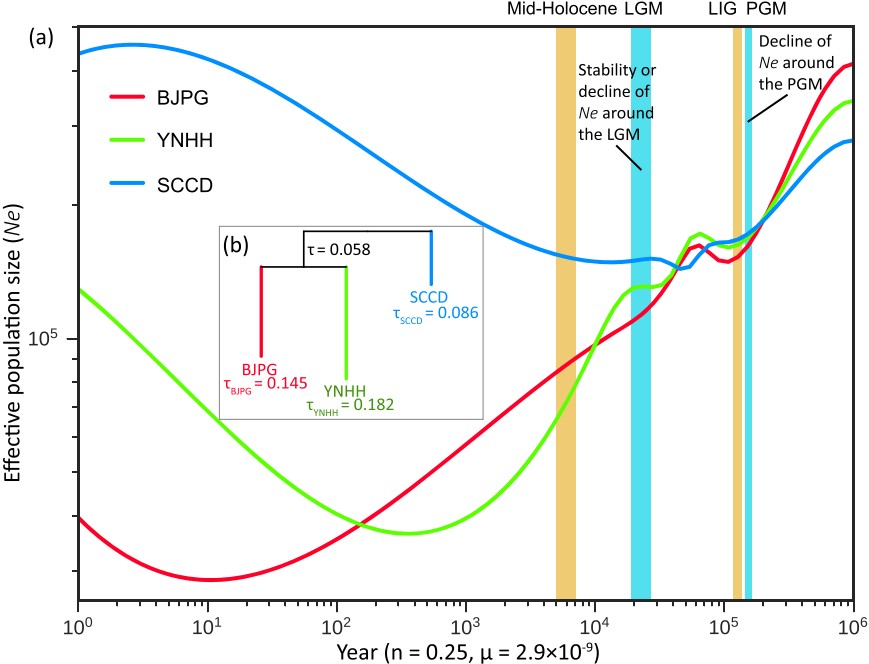

**Fig. 4 Variation in effective population size and divergence times of three populations of the oriental fruit moth. a** Variation in effective population size (*Ne*) inferred by the SMC++ analysis. Warm climatic periods are labeled by yellow bars, while cold climatic periods are labeled by blue bars. A mutation rate ($\mu$) of $2.9 \times 10^{-9}$ per site per generation and a generation time (*n*) of 0.25 years was used. **b** Divergence times estimated using KimTree. The parameter $\tau_k \equiv t_k/(2N_K)$ is the length of the branch leading to population k. Mid-Holocene, 6000 years before present; LGM the last glacial maximum, ~26.5–19 thousand years ago (kya); LIG the last interglacial period, 140–120 kya; PGM the penultimate glacial maximum, ~155–140 kya.

from Yunnan, and a more stable population size in another refugial population from Sichuan Basin during the Quaternary climatic oscillations. The high genomic differentiation among the two refugial populations and the lower differentiation between the YNHH and BJPG populations are consistent with the general pattern that the refugial populations usually persist with higher genetic diversity[54–56].

Even though the oriental fruit moth populations in the Sichuan Basin have higher genetic diversity, they have been restricted to their refugial areas, while lineages from the Yunnan region have expanded northward[12,32]. Larvae of the oriental fruit moth bore for long periods into fruit and tree shoots; they often disperse via the transport of fruit or nursery stock. During the last century, the oriental fruit moth has spread to almost every continent (reaching a cosmopolitan distribution) and shows high invasiveness[36]. Population genetic analyses showed that the globally dispersed populations are likely from other recently colonized populations rather than from the refugial area of Sichuan[12,30]. Thus, the SCCD population is likely to have remained relatively isolated. Molecular dating and demographic analysis has also shown that northerly populations of oriental fruit moth are derived from the refugial area of Yunnan in the Middle Pleistocene (about 54.0 kya) rather than Sichuan, whereas the two refugial populations of YNHH and SCCD diverged in the early Pleistocene (about 286 kya)[12,32,36]. Hence, strong local adaptation in the SCCD population was most likely shaped before the Middle Pleistocene.

A genome scan identified 42 genes on the neo-Z chromosome, including a circadian clock gene (*Clk*) and two target-sites of pesticides, as candidate genes associated with endemicity of the SCCD population. The *Clk* gene has been associated with breeding and migration in some birds[37,57–60]. According to our genome scanning methods, these genes have higher genetic differentiation between the SCCD and the other two populations, and less genetic diversity in the SCCD population. These results

indicate strong selection of these genes in the SCCD population. A previous study found that low genetic polymorphism at the *Clk* gene may limit adaptation to changing environments as suggested in other species[37,38]. It is well known that the oriental fruit moth overwinters by diapause of the last instar larvae[61]. Diapause induction of insects is associated with the circadian clock[62]. Perhaps range expansion of Sichuan populations is restricted by strong local adaptation of diapause induction. Further work is required to understand whether the *Clk* gene of the oriental fruit moth associates with diapause as well as with breeding and dispersal timing, and whether these relationships are sex-dependent, given that the *Clk* gene locates on the neo-Z chromosome.

Recent theory and empirical data suggest that selective processes can favor strong dispersal ability and result in the spatial sorting of genetic variation in dispersal ability[63–68]. In this study, several candidate genes and pathways involved in the development of wings were identified by exploring genome-wide selective sweeps in refugial and dispersed populations of the oriental fruit moth. Other genes potentially under selection include two amino sugar and nucleotide sugar metabolism genes, two *Cht3* genes, and two serpin genes. Amino sugar and nucleotide sugar metabolism are involved in the wing size of *Bombyx mori*[39]. *Cht3* is one of the insect chitinases and chitinase-like proteins from group III, whose orthologs regulate wing expansion and abdominal contraction in some insects[40–43]. *Serpin* genes represent a superfamily of protease inhibitors, whose ortholog was required for wing expansion in *Drosophila melanogaster*[44]. These candidates point to the possibility that the development of the oriental fruit moth wings may be under strong selection during range expansion. During the Quaternary climates, range expansion of species can last for a long period. Spatial sorting is the spatial analogue of natural selection that can lead to the adaptive divergence of populations in the center and range edge[69]. One of the common phenotypes under the selection of range expansion is dispersal ability[5,70]. Our results indicated the possibility of

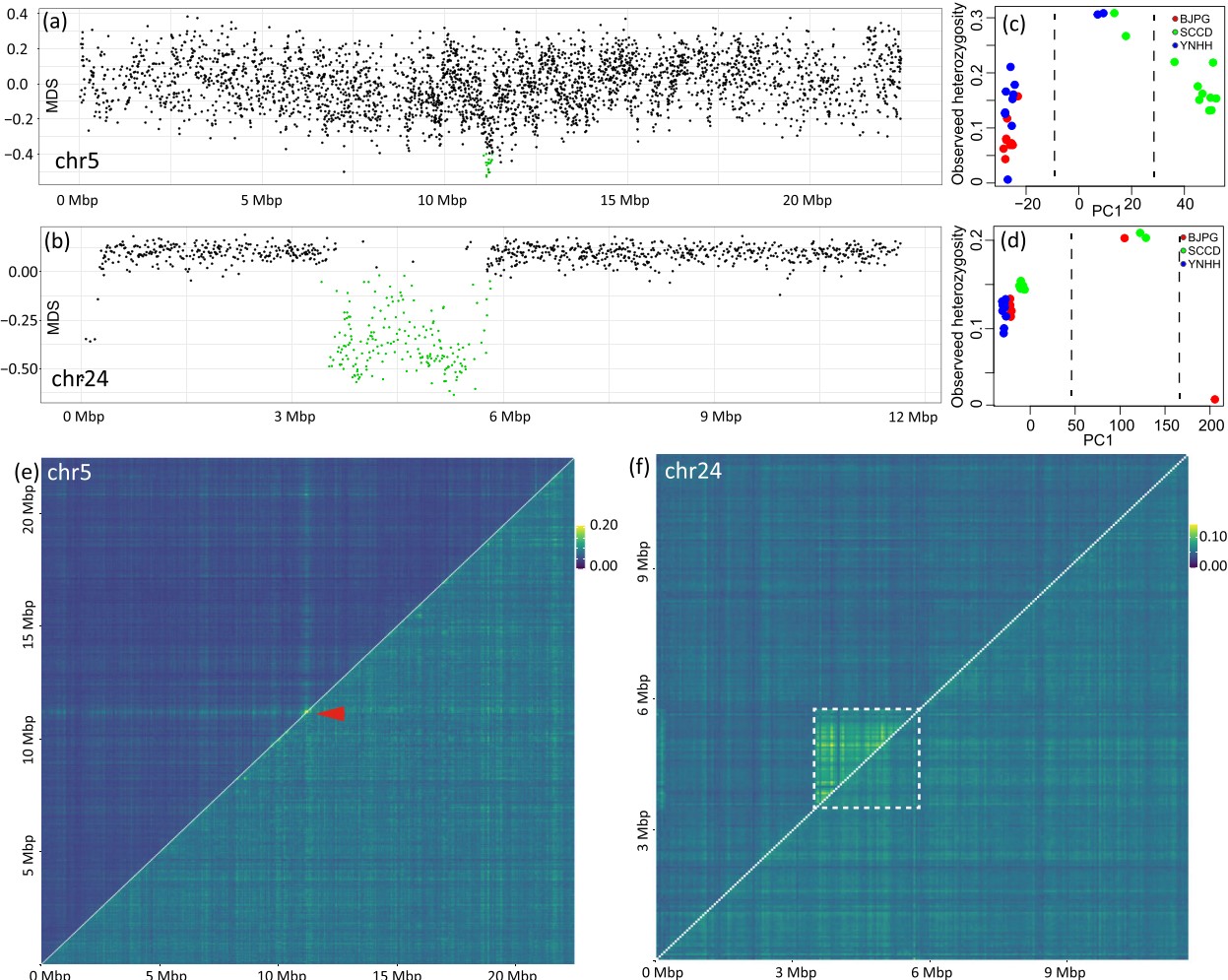

**Fig. 5 Genome inversions identified in the *Grapholita molesta*. a, b** Local PCA output for putative inversion region of chromosome chr5 and chr24. MDS, multidimensional scaling. Dots represent multidimensional scaling (MDS) for 100 SNP windows; outlier windows are highlighted in green. **c, d** Plots of the first component of a PCA and observed heterozygosity for the inversion regions of chromosome chr5 (**c**) and chr24 (**d**). **e, f** Heatmap of pairwise linkage disequilibrium for 50 kbp windows for chromosome chr5 (**e**) and chr24 (**f**); the upper triangle shows the linkage disequilibrium values for all individuals, while the lower triangle shows the linkage disequilibrium values when individuals have chromosomal inversion were excluded; the red arrow points to the small inversion region in chr5 (**e**), while the white box shows the large inversion region in chr24 (**f**).

spatial sorting in the oriental fruit moth during the Quaternary. Still, common garden experiments and gene function studies are needed to investigate this conjecture further.

Chromosome inversions could affect rates of adaptation, speciation, and the evolution of sex chromosomes[71,72]. This study reported two inversions in the oriental fruit moth for the first time to the best of our knowledge. The small inversion appeared differentiated between the refugial and the two other populations tested here, and its role in adaptive differentiation needs to be investigated.

## Methods

**Samples and library construction.** A laboratory-reared strain of the oriental fruit moth was used for de novo genome sequencing. This strain was derived from three male and female pairs and maintained on apples in laboratory conditions for ten generations. For population-level genotyping, we collected 263 individuals from 15 geographical populations across the native range of the oriental fruit moth in China. Four populations were collected from the Sichuan basin, four were from the Yunnan region, and seven were from regions to which the oriental fruit moth was subsequently dispersed. We chose three representative populations (31 individuals in total) for whole-genome resequencing. (Supplementary Table 5 and Fig. 3). The other 12 populations (232 individuals) were genotyped by the Kompetitive Allele-Specific PCR (KASP) method for 22 representative SNP outliers (see below).

We constructed and sequenced an Illumina library, a NanoPore library, a Hi-C proximity ligation library, and four RNA-seq libraries (eggs, larvae, pupae, and adults) for assembly and annotation of the oriental fruit moth genome. The raw reads generated from the Illumina platform were filtered by Trimmomatic v0.38[73] and then used to estimate genome size, heterozygosity, and duplication rate using GenomeScope v1.0[74].

**De novo genome assembly and annotation.** Long reads generated from the NanoPore platform were corrected and assembled using CANU version v1.8[75] with default parameters. The assembled contigs were polished based on Illumina short reads using Pilon v1.22[76]. To remove the possible secondary alleles, the assembled contigs were filtered using the pipeline Purge Haplotigs[77], resulting in a contig-level genome. The Illumina short reads sequenced from the Hi-C library were used to assemble these contigs into a chromosome-level genome using the Juicer v1.5[78] and 3D de novo assembly (3D-DNA) pipelines[79]. The completeness of each assembled version of the genome was assessed using a Benchmarking Universal Single-Copy Orthologs (BUSCO) v3.0.2[80] analysis, based on the insecta_odb9 database (1658 genes). We conducted a synteny analysis between the oriental fruit moth and the coding moth *C. pomonella* (Lepidoptera: Tortricidae) (Assembly accession: GCA_003425675.2)[29] and *Spodoptera litura* (Lepidoptera: Noctuidae) (Assembly accession: GCF_002706865.1)[81] using MCSCAN[82].

Repeats and transposable element families in the oriental fruit moth genome were detected by RepeatMasker pipeline v4.0.7[83] against the Insecta repeats within RepBase Update (http://www.girinst.org) and Dfam database (20170127), with RMBlast v2.10.0 as a search engine. tRNAs were annotated by tRNAscan-SE[84] with default parameters; rRNAs were annotated by RNAmmer prediction[85]. The

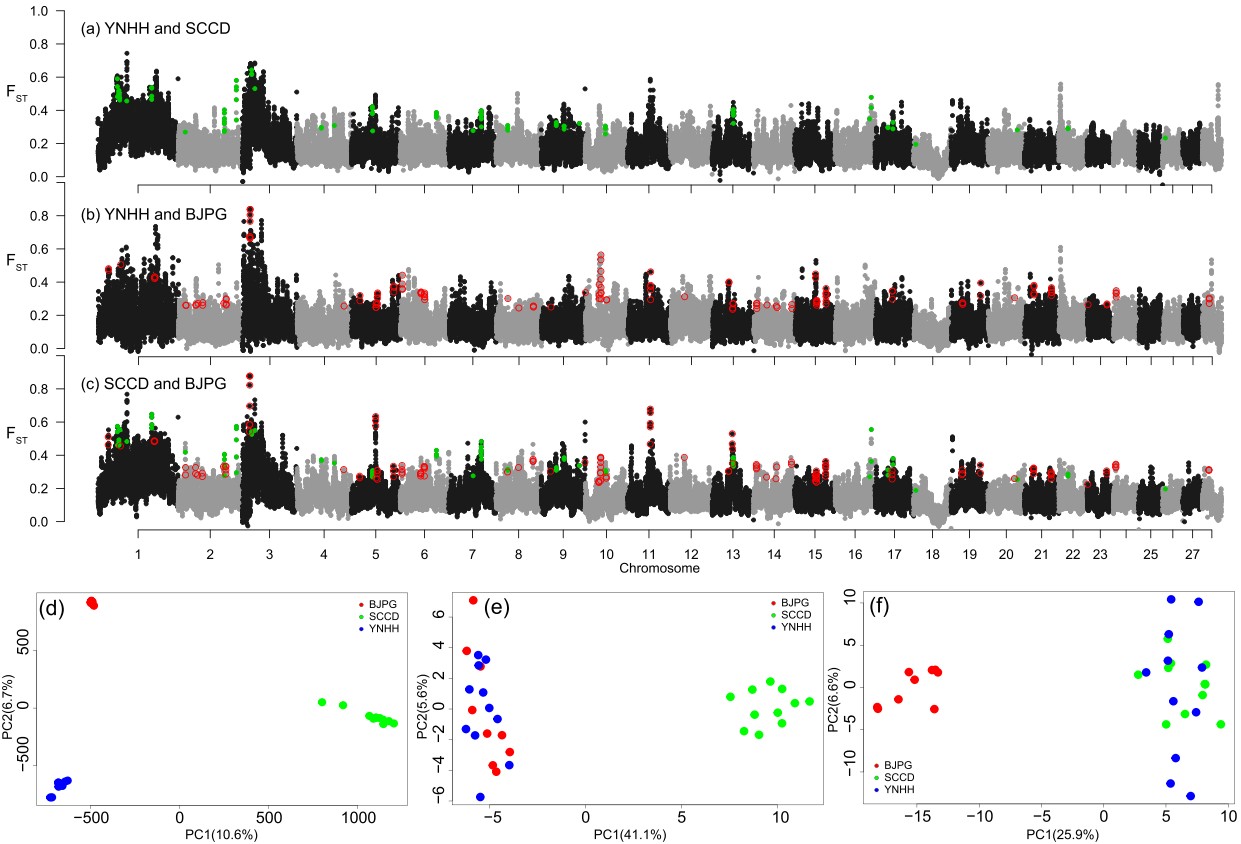

**Fig. 6 Identification of SNP outliers potentially related to the endemic SCCD and postglacial colonized BJYQ population of the *Grapholita molesta*.**
**a–c** The plot of pairwise $F_{ST}$ across 28 chromosomes for YNHH/SCCD (**a**), YNHH/BJPG (**b**), and SCCD/BJPG (**c**) population pairs. Green points indicate candidate regions related to the endemicity of the SCCD population, while red circles show candidate regions related to northward dispersal of the oriental fruit moth. **d-f** Scatter plots from the PCA analysis of individuals based on all SNPs (**d**), SNPs potentially related to the endemicity of the SCCD population (**e**), and SNPs potentially related to the northward dispersal of BJPG population (**f**).

protein-coding gene in the oriental fruit moth genome was annotated using ab initio, RNA-seq-based, and homolog-based methods in the MAKER version 3.01.03 genome annotation pipeline[86]. For the RNA-seq-based method, the RNA-seq reads were first mapped to the genome of the oriental fruit moth with Hisat v2.2.0. The transcripts were assembled using StringTie v2.1.2. For ab initio methods, parameters of SNAP v2013-02-16[87] and Augustus v3.2.3[88] were estimated or trained before using them to predict genes in MAKER. The SNAP parameters were estimated from high-quality transcripts obtained by improvement and filtering using PASA v2.4.1[89]. The gene model of Augustus was directly obtained from the above BUSCO analysis of the genome assembly. For the homolog-based method, protein-coding genes of *Drosophila melanogaster* and *Bombyx mori* were used. Fragments per kilobase per million (FPKM) values of each gene predicted by the MAKER pipeline were calculated using cufflinks version 2.2.1;[90] the gene set was filtered by keeping those with an FPKM value > 0 in any RNA-seq library. Finally, PASA was used to update the annotation based on transcripts; all predictions were further filtered using GffRead v0.11.7 implemented in Cufflinks v2.2.1[90] to remove genes having in-frame stop codons. Functions of the protein-coding genes were annotated using eggNOG-Mapper v1.0.3[91] against the database EggNOG v5.0[92]. Genes that can be functionally annotated by EGGNOG analysis were retained in gene structure annotation and used for further analysis.

**Genotyping populations of the oriental fruit moth across the native range.** Three representative populations were genotyped by genome resequencing, and 12 populations were genotyped using KASP (Fig. 2).

For genome resequencing of three representative populations, individual DNA libraries with an insert size of 400 bp were constructed and sequenced on the Illumina HiSeq X Ten platform to obtain 2 × 150 bp paired-end reads. A sequence depth of ~36-fold was obtained for each sample. After filtering out raw sequencing reads containing adapters and reads of low quality, the remaining clean reads were mapped to the reference assembly using BWA v0.7.17 with default parameters[93]. SAMtools v1.9[94] was used to sort reads and remove mapping quality lower than 30. Single-nucleotide polymorphism (SNP) calling was performed using the Genome Analysis Toolkit (GATK) v3.5[95]. The criteria used to filter the raw SNPs were

"QD < 2.0, FS > 60, SOR > 4.0, MQ < 40". SNPs were further filtered using the R package *vcfR*[96] and VCFtools v0.1.16[97] with the following criteria: SNPs with a sequencing depth lower than four and higher than 500 were removed; SNPs with a missing rate higher than 10% were removed. All the SNPs were annotated with SnpEff v4.3[98].

For genotyping the other 12 populations of the oriental fruit moth, we used the KASP, a fluorescence-based method. This method is suitable for detecting flexible numbers of SNPs with low cost and high accuracy. We selected 22 SNPs on nine outlier genes for genotyping (see below for outlier selection). Primers for each SNP were designed based on the genome sequence using SNP_Primer_Pipeline2 (https://github.com/pinbo/SNP_Primer_Pipeline2). The KASP reactions were conducted in 1,536 microplates with 1 μl final reaction volumes containing 1–5 ng genomic DNA, 0.5 μl of KASP 2x Master Mix (KBS-1016-011, LGC Genomics Ltd. Hoddesdon, UK), 0.5 μl of ultrapure water, 0.168 pmol of each allele-specific forward primer and 0.42 pmol reverse primer. The amplifications were conducted on an LGC Hydrocycler high-throughput thermal cycler (LGC Genomics Ltd. Hoddesdon, UK) using the following program: 95 °C for 15 min; 10 cycles of 94 °C for 20 s and touchdown starting at 61 °C for 60 s until to 55 °C, then 26–42 cycles of 94 °C for 20 s and 55 °C for 60 s. The fluorescence was scanned by Pherastar (LGC Genomics Ltd. Hoddesdon, UK), and the genotyping results were visualized and output files generated using Kraken software (LGC Genomics Ltd. Hoddesdon, UK). Due to unavailable KASP amplification of the Clk gene on the neo-Z chromosome, a pair of PCR primers (Clk_24750581_F, GCYAAWTATGCCAATTCCAA; Clk_24751090_R, ACTGTTTGCAGCGACCTACC) was designed for Sanger sequencing of the target region. PCR amplification and sequencing followed mitochondrial genes[12,32].

**Population genetic diversity and demographic analyses.** Genetic diversity and differentiation were estimated across chromosomes and among populations (Fig. 2). To characterize genetic diversity within and among populations, K-mers were first counted by jellyfish v 2.2.10 with a 17-base oligonucleotide. Genome size, heterozygosity, and duplication rate were then estimated for each individual using GenomeScope v1.0[74]. Pairwise genetic differentiation ($F_{ST}$) and nucleotide diversity ($\pi$) of the populations were calculated using VCFtools v0.1.16[97] with a window size

of 50 kb and widow step of 10 kb. The decay of LD against physical distance for the different populations was calculated using PopLDdecay[99] with default parameters.

Demography history were investigated in two perspectives (Fig. 2). The divergence times of the three populations was inferred using KimTree version 1.3[100] with the default Markov chain Monte Carlo (MCMC) parameters. We used SMC++ v1.15.4[101] to analyze the historical effective population size spanning $4 \times 10^6$ generations based on SNPs of autosomes. We assumed a mutation rate of $2.9 \times 10^{-9}$ per site per generation as estimated for *Heliconius* (Lepidoptera: Hesperiidae)[102], a generation time of 0.25 years[103,104].

**Genome structural variation analysis**. The program lostruct (local PCA/population structure, v.0.0.0.9) was used to detect chromosomal inversions from genome resequencing data[105]. Lostruct divides the genome into non-overlapping windows and calculates a PCA for each window. It then compares the PCAs derived from each window and calculates a similarity score. The matrix of similarity scores is then visualized using multidimensional scaling (MDS) transformation. Lostruct was run with 100 SNP-wide windows and independently for each chromosome. Each MDS axis was then visualized by plotting the MDS score against each window in the chromosome. We identified recombination suppression regions following[106]. Briefly, we manually selected the potential regions based on an MDS axis and minimum or maximum value that included windows within the region but excluded the rest of the chromosome. All SNPs within the regions defined by MDS scores were used to calculate PCAs using R to test whether they could divide samples into three groups representing 0/0, 0/1, and 1/1 genotypes. Average heterozygosity and LD within the regions were also calculated to test whether they were higher than the other chromosomes.

**Genome scan of genes potentially under selection**. We conducted genome scans to identify outliers potentially related to the refugial population and the northward dispersal (Fig. 2). The combination of the top 5% of $F_{ST}$ values and the top 5% of $\pi$ ratios between population pairs were considered selective sweep regions. Since $\pi$ and $F_{ST}$ values varied among chromosomes (see "Results"), each chromosome was analyzed independently. To detect outliers related to the northward dispersal of the oriental fruit moth, we used pairs of refuge/north populations, i.e., SCCD/BJPG and YNHH/BJPG. The intersection of these two combinations was considered as regions involving spatial sorting during northward dispersal. Missense SNPs with maf > 0.05 in these regions were extracted based on annotation results of SnpEff v4.3 analysis. These missense SNPs with higher $F_{ST}$ values between BJPG and the other two refuge populations than between YNHH and SCCD populations were retained as outlier SNPs. To detect outliers related to the endemicity of the SCCD population, we used $\pi$ ratios and $F_{ST}$ values of BJPG/SCCD and YNHH/SCCD and filtered outlier regions SNPs and genes as above.

We used Bedtools v2.2.80 (https://bedtools.readthedocs.io/en/latest/) to identify genes near candidate SNPs (±1 bp). We performed functional enrichment analyses using the clusterProfiler toolkit[107], where the significance level was set at 0.05, and the *P*-value was corrected using the Benjamini-Hochberg FDR.

**Reporting summary**. Further information on research design is available in the Nature Research Reporting Summary linked to this article.

## Data availability

The genome assembly has been deposited in the Genome repository (accession numbers: CP053120-CP053147) under NCBI BioProject PRJNA627114. The datasets used in data analysis are available in the Dryad repository (https://doi.org/10.5061/dryad.6wwpzgmzm) (ref. [108]).

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

## Acknowledgements

We thank Qiang Gao for his help in the assembly of Hi-C data. This research was supported by the National Natural Science Foundation of China (32070464), Joint Laboratory of Pest Control Research Between China and Australia (Z201100008320013), and Beijing Key Laboratory of Environmentally Friendly Pest Management on Northern Fruits (BZ0432).

## Author contributions

S.J.W. conceived and designed the study; L.J.C., W.S., and J.C.C. conducted the experiments; L.J.C., W.S., X.L.F., and S.J.W. analyzed the data; S.J.W., L.J.C., W.S., and A.H. discussed the results and wrote the manuscript.

## Competing interests

The authors declare no competing interests.
