## [Peer Review File · Communications Biology]

Reviewers' comments:

Reviewer #1 (Remarks to the Author):

The authors tried to investigate the impacts of the quaternary climate oscillations on the evolution of the oriental fruit moth with genomic data. It looks that the sequenced genomic data with new techniques are powerful, and the analyzing methods are persuasive and strong. However, I still have a few major concerns.

1. As far as I understand, it seems that the work done here is based on the hypotheses that the oriental fruit moth is native to East Asia with two refugial regions Yunnan and Sichuan Basin, and those from Yunnan expanded post-glacially to other areas, while those from Sichuan Basin remained endemically. Because the evolutionary history is invariably uncertain, I suggest the authors provide more evidence to justify this point, instead of only citing the papers you have published. In figure 1, it seems unnecessary to show the damage of the oriental fruit moth on fruits, and instead it is important to demonstrate its evolutionary routes.

2. I must admit that I cannot quite follow your analyzing methods of complexity. To make your methods useful for readers, especially for graduate students, I suggest the authors provide a flowchart to show your pipeline more clearly.

3. While you using the SMC++ to analyze the historical effective population sizes of the three geographic populations, did the authors analyze them separately or altogether? If altogether, is there a genealogy to show the relationship of the three populations? What is more, I am wondering how the genealogy of the 15 geographic populations looks like?

Minor points:

1. Many abbreviations in the text to confuse the understanding. Particularly, I don't think OFM is better than the moth, the fruit moth, or the oriental fruit moth.
2. L116: It is odd to find 'S.' first here, although the full genera name was given in Methods.
3. L332-334: How to understand 15 geographical populations, but three from Sichuan, three from Yunnan, and six from other regions?
4. L439: The fixation index F_{ST} is different from or the same as the F_{ST} in L413?
5. Fig.2: What do 'A' and 'G' mean? What do the subtitles mean? What are the colors in the pie for? Please state them more clearly.
6. Fig. 3: 'The left shadowed area indicates ... LGM, while the right shadowed area shows ... PGM'. The areas mean the blue bars?

Reviewer #2 (Remarks to the Author):

This study first published a new detailed genome for a pest moth species. The genome assembly took a benefit of several high-throughput platforms and I find this part done with an impressive precision. The genome was also annotated. A great benefit in this is that a genome for a close relative *C. pomonella* was available. While I am not aware of all bioinformatics details, it seems that those steps were also conducted in a very careful way. The same holds true for all methodological aspects of this work – all appear as well done.

The authors did not only create a new genome but used this genome to address biological questions. Particular attention was paid to understand how Quaternary climate oscillations shaped the genetic variability of the model species, and if traces of that are seen in the genomes of present-day specimens. I find this part not entirely clear. Particularly I find it problematic to understand the link between many of the conducted analyses and the addressed questions. I fully acknowledge that a summary of results of good to provide (lines 103-112), but for many of the next analyses (lines 113-159), it would be helpful for the reader to understand it better how these analyses link to the addressed questions. I even got a feeling that a lot of analyses were made without any clear purpose, but this may not be a fully correct idea, because some of these results were discussed – although not all. I would find it helpful for the reader the reasons for these

analyses being elaborated. Or, if no clear reason for the conducted analysis does not exist, I would consider leaving out some of the analyses. To exemplify my ideas, the authors state (from line 113): "The assembly of the OFM genome provides an opportunity to examine chromosome evolution...". This gave me an idea that this analysis was conducted because the data made it possible. But I find that the justification should come from the addressed questions. I would recommend going all the results through with this to be kept in mind.

I find the pictures clear and nice. Fig. 3 states that a generational time of 0.25 year was used. This sounds overly short given that most relatives have just a single generation per year. I remain wondering how much generation time being doubled or quadrupled here would have affected to the timing estimates.

Small issues:

Line 560: SuoMainen > Suomalainen

Line 551: (some problem here)

Line 644: Journal abbreviated here unlike elsewhere

Response to Editor and Reviewers' Comments

Reviewer #1 (Remarks to the Author):

The authors tried to investigate the impacts of the quaternary climate oscillations on the evolution of the oriental fruit moth with genomic data. It looks that the sequenced genomic data with new techniques are powerful, and the analyzing methods are persuasive and strong. However, I still have a few major concerns.

1. As far as I understand, it seems that the work done here is based on the hypotheses that the oriental fruit moth is native to East Asia with two refugial regions Yunnan and Sichuan Basin, and those from Yunnan expanded post-glacially to other areas, while those from Sichuan Basin remained endemically. Because the evolutionary history is invariably uncertain, I suggest the authors provide more evidence to justify this point, instead of only citing the papers you have published. In figure 1, it seems unnecessary to show the damage of the oriental fruit moth on fruits, and instead it is important to demonstrate its evolutionary routes.

>> **Response:** we have collected quite a lot of molecular data on this issue in the past and consider our papers the most relevant in tackling this question. However we have changed the wording in the introduction to provide some indication of caution in our interpretations. We removed the photo on damage.

2. I must admit that I cannot quite follow your analyzing methods of complexity. To make your methods useful for readers, especially for graduate students, I suggest the authors provide a flowchart to show your pipeline more clearly.

>> **Response:** We added a flowchart for the pipeline. We hope that this clarifies the analysis – please see Fig. 2.

3. While you using the SMC++ to analyze the historical effective population sizes of the three geographic populations, did the authors analyze them separately or altogether? If altogether, is there a genealogy to show the relationship of the three populations? What is more, I am wondering how the genealogy of the 15 geographic populations looks like?

>> **Response:** We analyzed them separately. The genealogy of three population was provided through analyzing their evolutionary history which is referred to in the text, while the genealogy of the 15 populations was not provided, because the population genomes of the other 12 geographic populations were not assessed. However we have completed other analyses in the past on multiple populations, please refer to previous papers.

Minor points:

1. Many abbreviations in the text to confuse the understanding. Particularly, I don't think OFM is better than the moth, the fruit moth, or the oriental fruit moth.

>> **Response:** Done.

2. L116: It is odd to find 'S.' first here, although the full genera name was given in Methods.

>> **Response:** Changed to *Spodoptera litura*.

3. L332-334: How to understand 15 geographical populations, but three from Sichuan, three from Yunnan, and six from other regions?

>> **Response:** It should be four from Sichuan, four from Yunnan, and seven from the other regions. Three from Sichuan, three from Yunnan, and six from the other regions were genotyped by the Kompetitive Allele-Specific PCR (KASP).

4. L439: The fixation index F_{ST} is different from or the same as the F_{ST} in L413?

>> **Response:** The description of this was deleted, as the analysis of F_{ST} values is now described earlier.

5. Fig.2: What do 'A' and 'G' mean? What do the subtitles mean? What are the colors in the pie for? Please state them more clearly.

>> **Response:** We added more details in the legends, and also provide more information on the subtitle and pie charts. Please note that this is now Fig. 3.

6. Fig. 3: 'The left shadowed area indicates ... LGM, while the right shadowed area shows ... PGM'. The areas mean the blue bars?

>> **Response:** We revised the description. Please see the legend at Fig. 4.

Reviewer #2 (Remarks to the Author):

This study first published a new detailed genome for a pest moth species. The genome assembly took a benefit of several high-throughput platforms and I find this part done with an impressive precision. The genome was also annotated. A great benefit in this is that a genome for a close relative *C. pomonella* was available. While I am not aware of all bioinformatics details, it seems that those steps were also conducted in a very careful way. The same holds true for all methodological aspects of this work – all appear as well done.

>> **Response:** Thanks for your positive comment.

The authors did not only create a new genome but used this genome to address biological questions. Particular attention was paid to understand how Quaternary climate oscillations shaped the genetic variability of the model species, and if traces of that are seen in the genomes of present-day specimens. I find this part not entirely clear. Particularly I find it problematic to understand the link between many of the conducted analyses and the addressed questions. I fully acknowledge that a summary of results of good to provide (lines 103-112), but for many of the next analyses (lines 113-159), it would be helpful for the reader to understand it better how these analyses link to the addressed questions. I even got a feeling that a lot of analyses were made without any clear purpose, but this may not be a fully correct idea, because some of these results were discussed – although not all. I would find it helpful for the reader the reasons for these analyses being elaborated. Or, if no clear reason for the conducted analysis does not exist, I would consider leaving out some of the analyses. To exemplify my ideas, the authors state (from line 113): “The assembly of the OFM genome provides an opportunity to examine chromosome evolution...”. This gave me an idea that this analysis was conducted because the data made it possible. But I find that the justification should come from the addressed questions. I would recommend going all the results through with this to be kept in mind.

>> **Response:** we added more details about the purpose of the different analyses. We maintained all the analyses which were included in the original paper because we felt that they provided useful insights into the different components of the work including population processes, chromosomal rearrangements and outlier loci. We now indicate more clearly why they were carried out. Please see the sentences added at the start of sections of the Results.

I find the pictures clear and nice. Fig. 3 states that a generational time of 0.25 year was used. This sounds overly short given that most relatives have just a single generation per year. I remain wondering how much generation time being doubled or quadrupled here would have affected to the timing estimates.

>> **Response:** In China, the oriental fruit moth has 3-6 generations per year which is why we used this figure. We have added the related references.

Small issues:

Line 560: SuoMainen > Suomalainen

>> **Response:** done.

Line 551: (some problem here)

>> **Response:** doi was added.

Line 644: Journal abbreviated here unlike elsewhere

>> **Response:** done.